# Vesicle Geometries Enabled by Semiflexible Polymer

**DOI:** 10.3390/polym14040757

**Published:** 2022-02-15

**Authors:** Ping Li, Nianqiang Kang, Aihua Chai, Dan Lu, Shuiping Luo, Zhiyong Yang

**Affiliations:** 1Department of Physics, Jiangxi Agricultural University, Nanchang 330045, China; lpfb1969@jxau.edu.cn (P.L.); mico19992000@163.com (N.K.); ludan@jxau.edu.cn (D.L.); luoshuipingjxau@163.com (S.L.); 2College of Data Science, Jiaxing University, Jiaxing 314001, China; ahchai@zjxu.edu.cn

**Keywords:** Monte Carlo method, semiflexible polymer, vesicle, vesicle shape

## Abstract

Understanding and controlling vesicle shapes is fundamental challenge in biophysics and materials design. In this paper, we employ the Monte Carlo method to investigate the shape of soft vesicle induced by semiflexible polymer outside in two dimensions. The effect of bending stiffness κ of polymer and the strength $\varepsilon_{VP}$ of attractive interaction between vesicle and polymer on the shape of vesicle is discussed in detail in the present paper. It is found that the shape of vesicle is influenced by κ and $\varepsilon_{VP}$. Typical shape of vesicles is observed, such as circular, cigar-like, double vesicle, and racquet-like. To engineer vesicle shape transformations is helpful for exploiting the richness of vesicle geometries for desired applications.

## 1. Introduction

Vesicles have existed since the first cells, and they are concerned with many biological processes and materials science, and they can be exploited for multitude of functionalizations in diverse fields, notably in DNA transfection of target cell, drug delivery [1,2], templating nanochemistry [3,4], micro-reactors [5], and the realization of relevant biological processes like endocytosis via the shape transformations of cell membrane [6]. Over the past few decades, a great deal of work about vesicles has been reported [7,8,9,10].

In biological systems, a large number of macromolecules, such as polysaccharides, membrane proteins, glococalix and cytoskeletons, decorate the extracellular part of the cell membrane. These “ornaments” are usually semiflexible or even rigid macromolecules, and the interactions between them and target cells bear an important interest, based on biomedical applications. An important advantage of synthetic polyanions is taken to protect mammalian hosts against a broad variety of viruses owing to the cytotoxic effect of the polyanion, e.g., copolymers of maleic anhydride, against both DNA and RNA pathogenic viruses [6,11]. In addition, they also play an important role in the other regulation of biological functions, such as signal transduction, endocytosis and exocytosis, cell motility, and cell mitosis.

Some researchers began to investigate responsive vesicles in the original basis of surfactant mixture systems via changing the external environments of vesicles, such as temperature [12] and PH [13]. Some researchers began to investigate responsive vesicles via changing the internal environments of vesicles, for example changing the concentration of polymers [14] and removing the water inside at different rates [15]. Yang et al. study the shape of vesicles which the rigid rods are anchored to, based on the self-consistent field theory [16]. Their results show that shape of vesicle is induced to change by the rod. However, subtle and complicated shape changes of membranes arising from the adsorption of polymer chains onto vesicle membranes have attracted considerable effort, based on theoretical approaches, computer simulations, and experiments [17,18,19,20,21,22,23,24]. Owing that the vesicle is impenetrable to polymer chains, the available space for the polymer decreases. Therefore, the inhomogeneous entropic pressure on the membrane is induced by the anchored polymer chains, and it give rise to the deformation of vesicle. As it was found that there exists adsorptive or repulsive interactions between the vesicle membrane and chain segments, the anchored polymers not only give rise to the inhomogeneous pressure on the membrane but also alter the local tension of the membrane in order to drive the vesicle deform remarkably. The local inhomogeneous curvature and bending rigidity of the membrane can be controlled by the polymer attached/adsorbed to the membrane [17,18,19,20,21,22,23,24]. The studies have suggested that polymer grafting leads to a stiffening of the membrane. The past studies pay more attention to the effect of flexible polymers on the vesicle shape. The influence of semiflexible polymer outside vesicle on the vesicle shape is studied by Monte Carlo method in the paper. The elasticity of vesicle is destroyed in the paper, therefore, the vesicle is flexible. The effect of attraction strength between polymer and vesicle, and rigidity of polymer on the vesicle shape is studied. The results presented here provide valuable insights to various biological processes, including cell motility, cell shape, and cell functions.

## 2. Model and Methods

In the paper, the Monte Carlo (MC) method is used to simulate phase transition process of vesicles induced by linear semiflexible polymer outside in the two-dimensional space. Vesicle can be considered as a ring polymer in two dimensions. Therefore, linear polymer and ring polymer composite can be used to simulate the interaction process of linear polymer and vesicle in two dimensions. Both the vesicle and the semiflexible polymer contain *N* + 1 effective monomers (where *N* is the chain length of semiflexible polymer), and their monomers are totally identical.

In the coarse-grained model of linear polymer, the potential of linear polymer is defined as the following form:(1)$$U_{linear}=U_{FENE}+U_{M}+U_{b}$$

The neighboring monomers are connected by the finitely extendable nonlinear elastic (FENE) potential [25,26]
(2)$$U_{FENE}=-kr_{0}{}^{2}\ln[1-{\left(\frac{l_{i}-l_{0}}{r_{0}}\right)}^{2}]$$

Here $l_{i}$ is *i*-th effective bond length, which alter in the range of $l_{\min}<l_{i}<l_{\max}$ (where *l*_min_ = 0.4 and *l*_max_ = 1.0), and its equilibrium distance *l*_0_ is 0.7 (where *l*_max_ is chosen to be the unit of length). $r_{0}=l_{\max}-l_{0}$, and the spring constant k is set to 20 in the units of *k_B_T* (where *k_B_* and *T* are the Boltzmann constant and the thermodynamic temperature, respectively). *k_B_T* is the unit of energy.

The excluded volume interaction among all non-bonded monomers is modeled by a Morse-type potential [25]
(3)$$U_{M}=\sum_{\left|i-j\right|>1}\varepsilon (\exp(-2\alpha (r_{ij}-r_{\min}))-2\exp(-\alpha (r_{ij}-r_{\min})))$$
where *r_ij_* is the distance between the *i*-th monomer and the *j*-th monomer, and we set α=24, *r*_min_ = 0.8, and ε=1, respectively. Because of the large α, *U_M_* decays to zero very rapidly for *r_ij_* > *r*_min_, and can be neglected completely for *r_ij_* > 1.0. The combination of FENE bonds with excluded volume interactions can prevent unphysical crossing of the polymer.

The bending energy which is used to describe the stiffness of polymer chain is modeled by an angle potential between adjacent bonds [25,26]
(4)$$U_{b}=\kappa (1+\cos\theta )$$
where θ is the bond angle, and κ is the bending stiffness. The chain rigidity can be altered by κ. Here, κ is in the units of *k_B_T*.

In the paper, the vesicle is soft. The potential of vesicle contains the finitely extendable nonlinear elastic (FENE) potential and the Morse-type potential, which are the same as the semiflexible polymer’s.

The interaction between vesicle and linear polymer chain is via Lennard-Jones potential [27] (5)$$U_{LJ}=4\varepsilon_{VP}[{\left(\frac{\sigma }{r}\right)}^{12}-2\left(\frac{\sigma }{r}{)}^{6}\right]$$ where *r* is the distance between the monomer of vesicle and the monomer of linear polymer chain. The energy parameter $\varepsilon_{VP}$ is used to adjust the attractive strength between vesicle and linear polymer chain.

All results are acquired by the off-lattice Monte Carlo (MC) simulations [28]. MC simulations are carried out based on the Metropolis algorithm, and MC method is extensively used to study phase transition of polymer chains at low temperature or strong attractive interactions [29] because some complex transition behavior should be studied by sophisticated computer simulation methodologies [30], such as generalized ensemble Monte Carlo simulation algorithms such as multicanonical sampling [31] or the Wang–Landau method [32]. In detail, for each trial move, a monomer is randomly chosen and trys to move from its position (x_0_, y_0_) to a new position (x, y) with increments Δx and Δy, which are selected randomly from the intervals (−0.25,0.25), respectively. The trial move is accepted if Δ > η, where η = min(exp (−ΔU/*k*_B_*T*),1) is the transition probability depending on the difference in energy ΔU between the trial and old states and Δ is chosen from the interval [0, 1] by the random function. The time unit is one Monte Carlo step (MCS) during which 2(*N* + 1) moves are tried. Direct simulation of phase transition faces the obstacle of high free energy barriers. At phase transition points, metadynamics method is employed here to surmount the free energy barrier. The free energy profile is calculated to determine the accurate transition points [33,34]. To ensure the composite reach equilibrium, $3.0\times 10^{8}$ MCSs are performed. The data are collected by averaging over 100 independent runs. The initial conformation of the system is different, and seed of random generator is different for each independent run. We measure once every $1.0\times 10^{6}$ MCSs, and 100 measurements are performed in each independent run. Therefore, each statistical quantity is averaged over 10,000 samples. Their errors are less than symbol, therefore they are not shown in the Figures. We set chain length *N* = 60.

## 3. Results and Discussion

### 3.1. Effect of Bending Stiffness of Polymer on the Shape of Vesicles

For semiflexible polymers, the mechanical properties are well described by a self-avoiding walk [35]. Kuhn length is changed to vary the rigidity of biopolymer [36]. The snapshots of Figure 1 show that rigidity of linear polymer has very obvious effect on the shape of vesicles. Figure 1a shows that the shape of vesicle is two-fold asymmetry at κ = 10, i.e., it forms asymmetrical double vesicle; its shape is circular at κ = 50 and 100; its shape becomes oval at κ = 150; its shape is random loop at κ ≥ 200 for $\varepsilon_{VP}$ = 2. As κ is small, the polymer is flexible. The entropy takes the dominate role and the attractive polymer-vesicle interaction is weak, thus, the conformation of vesicle is disorder. Double vesicle is helpful to decrease the attractive energy so as to reduce the free energy of the system. As κ is moderate, polymer is semiflexible. The bending energy of polymer takes important role. The competition between the bending energy and entropy is main. The polymer folding into loop is helpful to decrease the bending energy so as to decrease the free energy to minimum. As κ is large, polymer is rigid. For small $\varepsilon_{VP}$, the bending energy plays the dominate role. The polymer becomes rod-like, therefore, only part of vesicle adsorbs onto the polymer. Figure 1b shows that the vesicle has two-fold asymmetry firstly, then has two-fold symmetry, finally it has two-fold asymmetry again with κ increase from 10 to 100 for $\varepsilon_{VP}$ = 8; its shape is cigar-like at κ = 150; with κ increase further, its shape becomes circular. For $\varepsilon_{VP}$ = 8, the attractive polymer-vesicle interaction is moderate. As κ is not very large, the attractive interaction plays dominate role. Two-fold is helpful to decrease the attractive energy so as to reduce the free energy to minimum. The role of bending energy becomes more and more important with κ increase, and it becomes more and more difficulty for polymer to bend. Therefore, the cigar-like shape is determined by the balance between attractive interaction and bending energy for κ = 150. Even if κ = 200 or 300, the bending energy is not rigid enough for polymer to separate from the vesicle. The vesicle becomes circular. It is helpful to reduce the bending energy so as to decrease the free energy to minimum. Figure 1c shows that the vesicle has two-fold symmetry at κ = 10, 50, and 100, its shape becomes bent-cigar-like at κ = 150 and 200, its shape becomes racquet-like at κ = 300 for $\varepsilon_{VP}$ = 12. The results are in agreement with recent simulation results on the structure formation (in two-dimensions) in colloidal particles [37]. The shape of vesicle is determined by the balance among the bending energy of semiflexible polymer, attractive energy between linear polymer chain and vesicle, and entropic pressure exerted by the semiflexible polymer. The results of de la Cruz et al. show that the vesicle shape can be cigar-like, whistle-like, double vesicle or sphere-like by removing water from the vesicle at different rates [16]. The study results of Yang et al. show that vesicle anchored by rigid rod or polymer has two-fold symmetry at certain condition [16,38]. Our results are in agreement with theirs.

The average attractive polymer-vesicle energy <*U_VL_*> is discussed. <*U_VL_*> increases from −53.83 to −47.19 with κ increase from 10 to 50, firstly, then almost keeps constant in the range of κ = 50~150, then increases with κ increase from 150 to 200, finally keeps constant with κ increase for $\varepsilon_{VP}$ = 2, as shown in Figure 2. It indicates that the vesicle undergoes two phase transition with κ increase. It is in agreement with the results of Figure 1. For $\varepsilon_{VP}$ = 8, <*U_VL_*> has a little fluctuation in the range of κ = 10~100, then increases with κ increase from 100 to 200, finally keeps constant in the range of κ = 200~300. It indicates that there is an intermediate shape with κ increase. For $\varepsilon_{VP}$ = 12, <*U_VL_*> slightly increases with κ increase in the range of κ = 10~100 firstly, then increases quickly with κ in range of κ = 100~150, finally begins to increase slightly again with κ increase in the range of κ = 150~300. It indicates that the vesicle shape at κ = 10~100 is different from those at κ = 150~300. The results of mean end-to-end distance <*R_ee_*> of semiflexible polymer are in agreement with the results of Figure 2, as shown in Figure 3.

Next, the vesicle shape is studied by the tangent–tangent correlation <***b_k_***·***b_k+s_***> [39,40] which is an ideal observable reflecting the statistics along the whole contour of the two-dimensional vesicle (where ***b_k_*** is the *k*th bond vector), as shown in Figure 4. Figure 4a shows <***b_k_***·***b_k+s_***> at different κ for $\varepsilon_{VP}$ = 2. <***b_k_***·***b_k+s_***> presents periodicity for κ = 10, and the number of periods is in agreement with the number of folds; it is perfect cosnusoid for κ = 50, 100 and 150; it presents a perfect “V” for κ = 200 and 300. It indicates that the vesicle has two-fold for κ = 10, the shape of vesicle is circular for κ = 50, 100 and 150, a part of vesicle adsorbs onto rigid polymer for κ = 200 and 300. Figure 4b shows <*b_k_*·*b_k+s_*> at different κ for $\varepsilon_{VP}$ = 8. <*b_k_*·*b_k+s_*> decreases linearly in the range of *s* < 31, and it slightly increases linearly in the range of *s* = 31–51, then increases sharply in the range of *s* > 51 for κ = 150. It indicates that shape of vesicle is cigar-like. In addition, the results of <*b_k_*·*b_k+s_*> indicate that the vesicle forms double vesicle for κ = 10, 50, and 100, and the shape of vesicle is circular for κ = 200 and 300. Figure 4c shows <*b_k_*·*b_k+s_*> at different κ for $\varepsilon_{VP}$ = 12. The results of <*b_k_*·*b_k+s_*> at κ = 10, 50 and 100 indicate that the vesicle forms double vesicle. The curve of κ = 300 is different from the curve of κ = 150 and 200, The main reason is that shape of vesicle is bent-cigar-like for κ = 150 and 200, while it is racquet-like for κ = 300.

### 3.2. Effect of on the Shape of Vesicles 

The effect of attractive polymer-vesicle interaction on the vesicle shape is studied, as shown in Figure 5. For κ = 10, the vesicle has a two-fold asymmetry for $\varepsilon_{VP}$ = 2; vesicle has three-fold asymmetry with $\varepsilon_{VP}$ increase to 4; with $\varepsilon_{VP}$ increase further, the vesicle becomes double vesicle again and the double vesicle becomes more and more symmetrical with ε increase from $\varepsilon_{VP}$ = 6 to $\varepsilon_{VP}$ = 12, as shown in Figure 5a. For κ = 150, the shape of vesicle is oval at $\varepsilon_{VP}$ = 2 and $\varepsilon_{VP}$ = 4, it becomes racquet-like at $\varepsilon_{VP}$ = 6, and finally becomes cigar-like at $\varepsilon_{VP}$ = 8 and becomes bent-cigar-like with $\varepsilon_{VP}$ increase further, as shown in Figure 5b. For κ = 300, only partial vesicle adsorbs onto polymer at small $\varepsilon_{VP}$, the shape of vesicle shifts from oblate oval to swelled oval with $\varepsilon_{VP}$ increase from $\varepsilon_{VP}$ = 4 to $\varepsilon_{VP}$ = 8; the shape of vesicle becomes racquet-like at $\varepsilon_{VP}$ = 12.

Figure 6 shows that the average attractive polymer-vesicle energy <*U_VL_*> decreases with $\varepsilon_{VP}$ increase. <*U_VL_*> of κ = 300 is more than twice that of κ = 10 and 150 at $\varepsilon_{VP}$ = 2. It means that only partial polymer adsorbs onto vesicle. The slope of κ = 10 is larger than that of κ = 150 and 300. It indicates that the conformation of vesicle for κ = 10 is more compact than that for κ = 150 and 300 at any $\varepsilon_{VP}$. To understand the shape transformation further, mean end-to-end distance <*R_ee_*> of semiflexible polymer is studied, as shown in Figure 7. For κ = 10, <*R_ee_*> decreases firstly and reaches minimum at $\varepsilon_{VP}$ = 4.0, then increases, finally it almost keeps constant with ε increase. It indicates that vesicle shape of $\varepsilon_{VP}$ = 4 is different from that of other $\varepsilon_{VP}$, and the shape of vesicle adjusts slightly for $\varepsilon_{VP}$ ≥ 6. The results of κ = 150 indicate that the vesicle shape of $\varepsilon_{VP}$ = 2~4 is different from that of $\varepsilon_{VP}$ = 6~12. For κ = 300, <*R_ee_*> is about 39.40 at $\varepsilon_{VP}$ = 2. The value is almost equal to the length that the polymer totally extends. It means that only partial vesicle adsorbs onto polymer. <*R_ee_*> fluctuates around 2.0 in the range of $\varepsilon_{VP}$ = 4~12. It indicates that the vesicle is totally enveloped by polymer.

Next, the vesicle shape is studied by the tangent–tangent correlation <*b_k_*·*b_k+s_*>. Figure 8a shows that <*b_k_*·*b_k+s_*> of $\varepsilon_{VP}$ = 4 has three troughs, while <*b_k_*·*b_k+s_*> of other ε has two troughs for κ = 10. It indicates that the vesicle forms triple vesicle for $\varepsilon_{VP}$ = 4, while it forms double vesicle for other $\varepsilon_{VP}$. The results of Figure 8b indicates that vesicle shape of $\varepsilon_{VP}$ = 2, 4 and 6 is circular, the vesicle shape of $\varepsilon_{VP}$ = 8 is cigar-like, the vesicle shape of $\varepsilon_{VP}$ = 12 is bent-cigar-like for κ = 150. The results of Figure 8c indicate that the vesicle shape firstly shifts from a random shape to a circular shape, then shifts to a racquet-like shape with ε increase for κ = 300.

## 4. Conclusions

The shape transition of vesicle induced by semiflexible polymer outside is studied by the off-lattice Monte Carlo method in two dimensions. We discuss the effect of κ and $\varepsilon_{VP}$ on the vesicle shape in detail. Typical vesicle shapes are observed. When $\varepsilon_{VP}$ is small, the vesicle firstly transforms from asymmetrical double vesicle to circular vesicle, then shifts to random vesicle with κ increase. When $\varepsilon_{VP}$ is moderate, the vesicle transforms from double vesicle to cigar-like vesicle, then becomes circular vesicle with κ increase. When $\varepsilon_{VP}$ is large, the vesicle transforms from double vesicle to bent-cigar-like vesicle firstly, then becomes racquet-like vesicle with κ increase. When the rigidity of polymer is weak, the vesicle shape is the double vesicle for all $\varepsilon_{VP}$. When κ increases, more types of vesicle shape are observed. For moderate κ, vesicle transforms from circular vesicle to racquet-like vesicle, then becomes cigar-like vesicle with κ increase. For large κ, shape of vesicle is circular at moderate $\varepsilon_{VP}$, while it becomes racquet-like at large $\varepsilon_{VP}$. The results presented here provide valuable insights to various biological processes, including cell motility, cell shape, and cell functions.

## Figures and Tables

**Figure 1 polymers-14-00757-f001:**
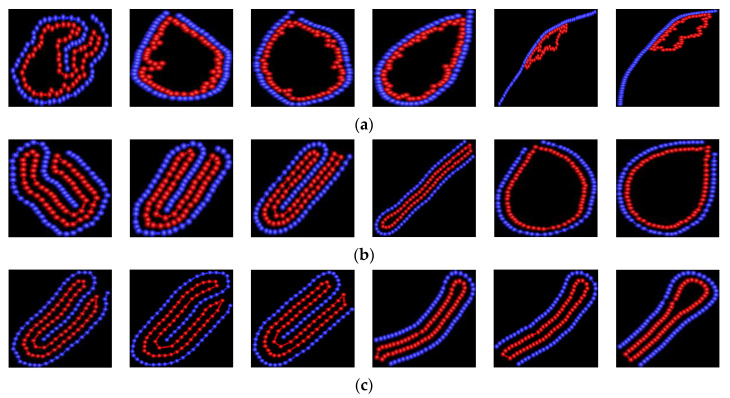
The snapshots of semiflexible polymer/vesicle composites. The blue one represents the linear polymer, and the red one represents vesicle. In the snapshots, the bending stiffness of polymer is 10, 50 100, 150, 200, and 300 from left to right for (**a**) $\varepsilon_{VP}$ = 2, (**b**) $\varepsilon_{VP}$ = 8, and (**c**) $\varepsilon_{VP}$ = 12.

**Figure 2 polymers-14-00757-f002:**
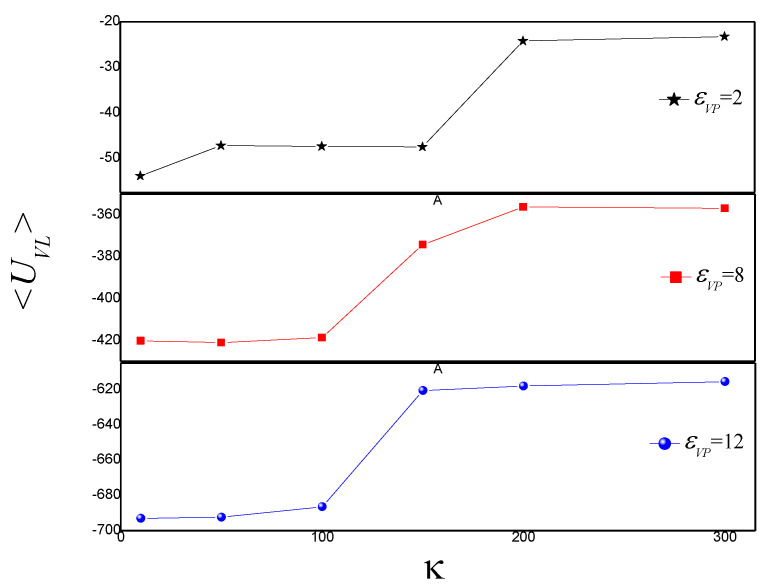
The average attractive energy <*U_VL_*> between the semiflexible polymer chain and vesicle versus bending stiffness κ for $\varepsilon_{VP}$ = 2, 8, and 12.

**Figure 3 polymers-14-00757-f003:**
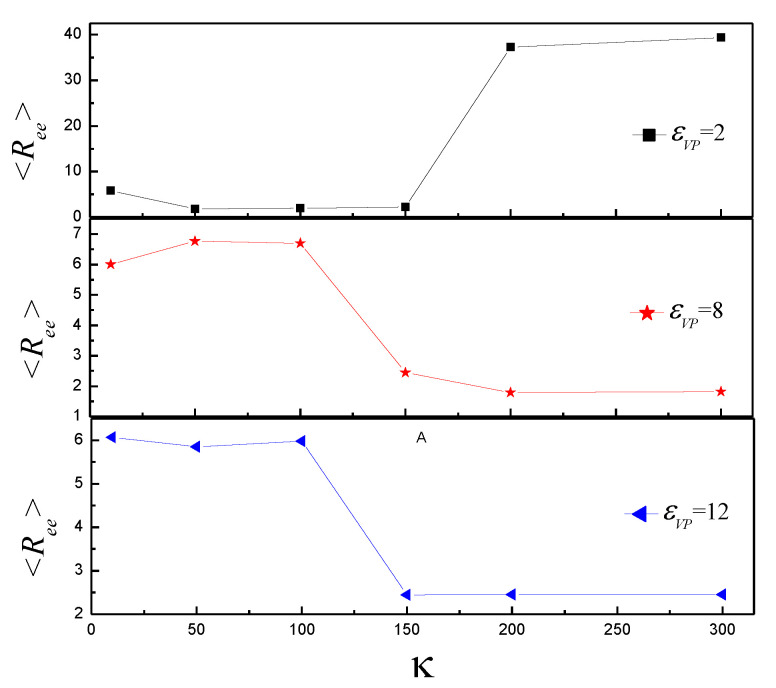
The mean end-to-end distance <*R_ee_*> versus bending stiffness κ for $\varepsilon_{VP}$ = 2, 8, and 12.

**Figure 4 polymers-14-00757-f004:**
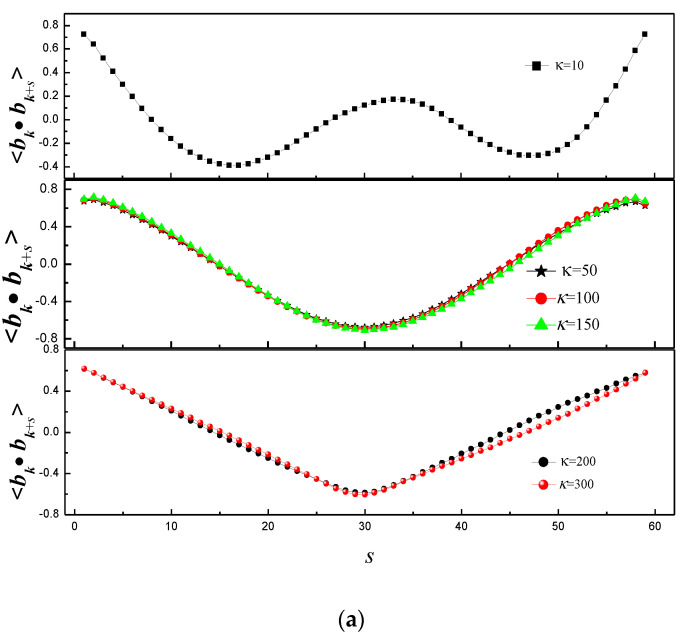
Tangent–tangent correlation of vesicle whose conformation is induced by linear polymer of different rigidities for (**a**) $\varepsilon_{VP}$ = 2, (**b**) $\varepsilon_{VP}$ = 8, and (**c**) $\varepsilon_{VP}$ = 12.

**Figure 5 polymers-14-00757-f005:**
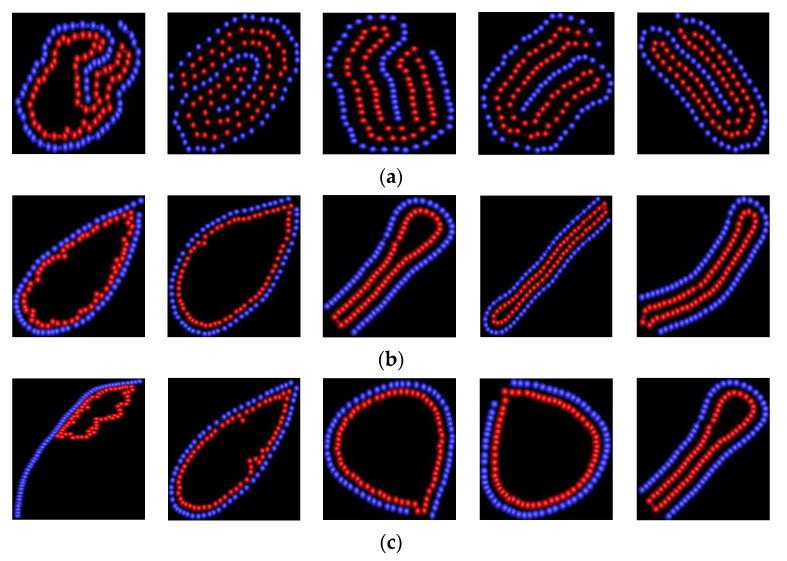
The snapshot of linear semiflexible polymer/ring polymer composites. The blue one represents the linear polymer, and the red one represents vesicle. The of snapshots is 2, 4, 6, 8, and 12 from left to right for (**a**) κ = 10, (**b**) κ = 150, and (**c**) κ = 300.

**Figure 6 polymers-14-00757-f006:**
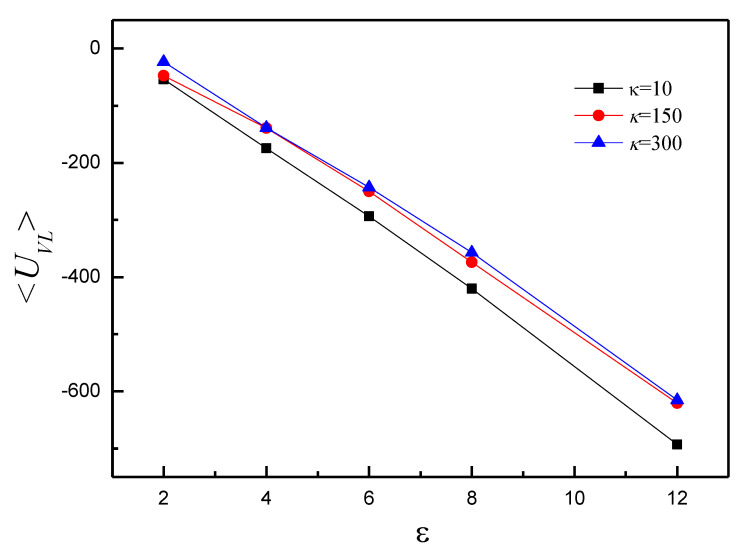
The average attractive energy <*U_VL_*> between the semiflexible polymer chain and vesicle versus $\varepsilon_{VP}$ for κ = 10, 150, and 300.

**Figure 7 polymers-14-00757-f007:**
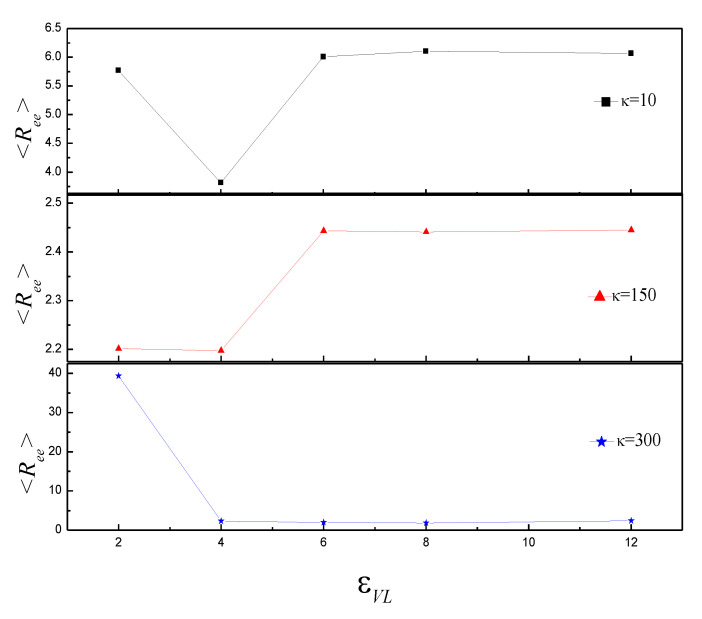
The mean end-to-end distance <*R_ee_*> versus $\varepsilon_{VP}$ for κ = 10, 150, and 300.

**Figure 8 polymers-14-00757-f008:**
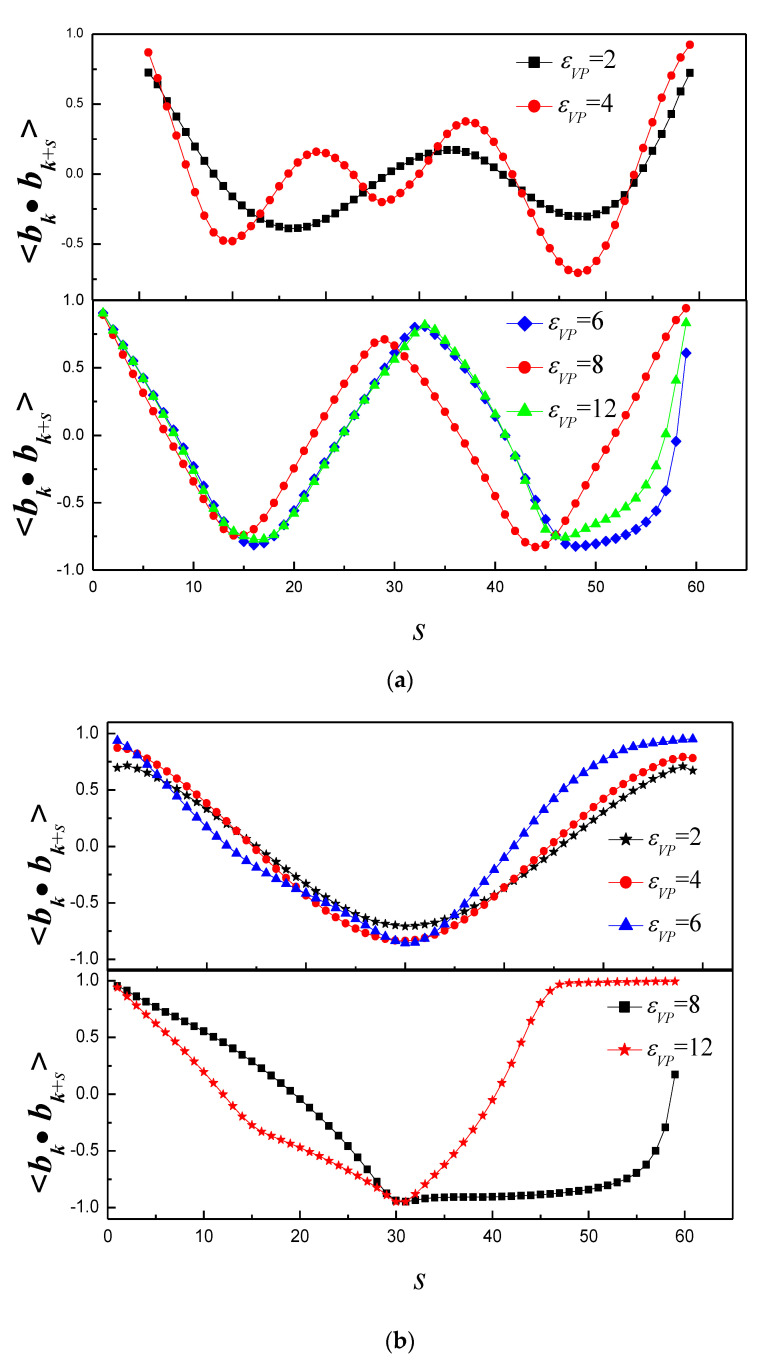
Tangent–tangent correlation of vesicle whose transformation is induced by linear polymer of (**a**) κ = 10, (**b**) κ = 150, and (**c**) κ = 300.

## Data Availability

The data presented in this study are available on request from the corresponding author.

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
