# Peer review of "Vesicle Geometries Enabled by Semiflexible Polymer"

_polymers, 2022, doi:10.3390/polym14040757_

Round 1
Reviewer 1 Report
Dear Colleagues,
In this review article, the authors employ the Monte Carlo method to investigate the shape of soft vesicle induced by semiflexible polymer outside in two dimensions. The applied method is described in the lines 98-110 of the article. It uses a random number generator. More correctly, a pseudo-random number generator. Since the generation of pseudo-random numbers occurs a huge number of times, repeatability can appear in the resulting sequences of numbers. I believe that the authors need to indicate this in the article, and check the results obtained on different pseudo-random number generators (or with different pseudo-random number generator settings). I recommend publishing the review article "Vesicle Geometries Enabled by Semiflexible Polymer" when correcting these question.
Best regards,
Reviewer.
Author Response
In this review article, the authors employ the Monte Carlo method to investigate the shape of soft vesicle induced by semiflexible polymer outside in two dimensions. The applied method is described in the lines 98-110 of the article. It uses a random number generator. More correctly, a pseudo-random number generator. Since the generation of pseudo-random numbers occurs a huge number of times, repeatability can appear in the resulting sequences of numbers. I believe that the authors need to indicate this in the article, and check the results obtained on different pseudo-random number generators (or with different pseudo-random number generator settings). I recommend publishing the review article "Vesicle Geometries Enabled by Semiflexible Polymer" when correcting these question.
Answer: The resulting sequences of numbers is different for different seed of random function on our pseudo-random number generators. Therefore, we set different seed of random function in each different independent run. In addition, if the initial conformation of system is different for each run, even if the resulting sequences of numbers are same, the results are different. We have stated them in the revised manuscript.
Reviewer 2 Report
In this paper, Li et al. have done MC simulations to investigate the effects of bending stiffness and magnitude of polymer (linear semiflexible polymer surrounding the vesicle, modeled as a ring polymer chain) vesicle interactions on the shape of vesicle. Possible two-dimensional phase transitions are studied and the structured involved are discussed. I recommend publication of the paper, subject to the following major revisions:
1-The phases involved in vesicular systems have been studied using unbiased (brute force) simulations. As the phase transition is a slow process, the free-energy barrier involved in the transition, cannot be surmounted in the time scale of simulations. This means that large perturbations (strength of polymer-vesicle interactions or bending stiffness) cause the phase transition in the time scale of simulations. However, in this case the phase transition points cannot be calculated accurately. To calculate the accurate transition points, one should invoke advanced sampling simulations (free energy calculations, see for example J. Chem. Theory Comput. 2019, 15, 1345, J. Phys. Chem. B 2020, 124, 10374). I am fine with the type of simulations done in this work, but I think a few sentences in this respect should be included in the text.
2-Figures 1, 2 and 6: Increasing the epsilon values causes the phase transition from asymmetrical double vesicles to racquet-like structures (more compact structures) in two-dimensions. These results are in agreement with recent simulation results on the structure formation (in two-dimensions) in colloidal particles (see for example J. Chem. Theory Comput. 2021, 17, 1742). An interesting point consistent with the calculations in the cited paper (employing free energy calculation methods) is that the structures evolve from disordered structures (at low epsilon) to ring structures (at higher epsilon values) to close packed structures (at quite high epsilon values). Perhaps giving address to this point helps the readers to follow the literature on a closely related topic.
3-Minor point: In some places in the text epsilons are written as epsilon, but in some other places as (epsilon)vp. Please make them all consistent.
Author Response
1-The phases involved in vesicular systems have been studied using unbiased (brute force) simulations. As the phase transition is a slow process, the free-energy barrier involved in the transition, cannot be surmounted in the time scale of simulations. This means that large perturbations (strength of polymer-vesicle interactions or bending stiffness) cause the phase transition in the time scale of simulations. However, in this case the phase transition points cannot be calculated accurately. To calculate the accurate transition points, one should invoke advanced sampling simulations (free energy calculations, see for example J. Chem. Theory Comput. 2019, 15, 1345, J. Phys. Chem. B 2020, 124, 10374). I am fine with the type of simulations done in this work, but I think a few sentences in this respect should be included in the text.
Answer: We have added them in the revised manuscript.
2-Figures 1, 2 and 6: Increasing the epsilon values causes the phase transition from asymmetrical double vesicles to racquet-like structures (more compact structures) in two-dimensions. These results are in agreement with recent simulation results on the structure formation (in two-dimensions) in colloidal particles (see for example J. Chem. Theory Comput. 2021, 17, 1742). An interesting point consistent with the calculations in the cited paper (employing free energy calculation methods) is that the structures evolve from disordered structures (at low epsilon) to ring structures (at higher epsilon values) to close packed structures (at quite high epsilon values). Perhaps giving address to this point helps the readers to follow the literature on a closely related topic.
Answer: We have given address in the revised manuscript.
3-Minor point: In some places in the text epsilons are written as epsilon, but in some other places as (epsilon)vp. Please make them all consistent.
Answer: We have revised them in the revised manuscript
Round 2
Reviewer 2 Report
The revised version of manuscript is improved substantially. I recommend publication.